# HyResPINNs: Adaptive Hybrid Residual Networks for Learning Optimal Combinations of Neural and RBF Components for Physics-Informed Modeling

## Abstract

Physics-informed neural networks (PINNs) are an increasingly popular class of techniques for the numerical solution of partial differential equations (PDEs), where neural networks are trained using loss functions regularized by relevant PDE terms to enforce physical constraints. We present a new class of PINNs called HyResPINNs, which augment traditional PINNs with adaptive hybrid residual blocks that combine the outputs of a standard neural network and a radial basis function (RBF) network. A key feature of our method is the inclusion of adaptive combination parameters within each residual block, which dynamically learn to weigh the contributions of the neural network and RBF network outputs. Additionally, adaptive connections between residual blocks allow for flexible information flow throughout the network. We show that HyResPINNs are more robust to training point locations and neural network architectures than traditional PINNs. Moreover, HyResPINNs offer orders of magnitude greater accuracy than competing methods on certain problems, with only modest increases in training costs. We demonstrate the strengths of our approach on challenging PDEs, including the Allen-Cahn equation and the Darcy-Flow equation. Our results suggest that HyResPINNs effectively bridge the gap between traditional numerical methods and modern machine learning-based solvers.

## 1 Introduction

Partial differential equations (PDEs) model a variety of phenomena across science and engineering and are traditionally solved using numerical methods such as finite difference methods (LeVeque, 2007) and finite elements (Strang et al., 1974). Physics-informed neural networks (PINNs) are meshless methods (Raissi et al., 2019; Raissi, 2018) that solve PDEs by training deep feedforward neural networks (DNNs) using PDEs as soft constraints. This traditional PINN training method poses challenges due to complicated loss landscapes arising from the PDE-based soft constraints (Krishnapriyan et al., 2021). Recent approaches to ameliorating issues include curriculum learning (Krishnapriyan et al., 2021), novel optimization techniques (Cyr et al., 2020), domain decomposition (X-PINNs) (Jagtap & Karniadakis, 2020), gradient-enhanced training (G-PINNs) (Yu et al., 2022a), or discretely-trained PINNs using RBF-FD approximations in place of automatic differentiation (DT-PINNs) (Sharma & Shankar, 2022). Further significant strides have been made in developing new DNN architectures to enhance PINNs' representational capacity, including adaptive activation functions Jagtap et al. (2020b;a), positional embeddings Liu et al. (2020); Wang et al. (2021b), and innovative architectures Wang et al. (2021a); Sitzmann et al. (2020); Gao et al. (2021); Fathony et al. (2020); Moseley et al. (2023); Kang et al. (2023).

In addition to architectural choices, certain training pathologies exist including, spectral bias (Rahaman et al., 2019; Wang et al., 2021b), unbalanced back-propagated gradients (Wang et al., 2021a; 2022b), and causality violations (Wang et al., 2022a; 2024b). Efforts focused on improving PINNs' training performance include loss re-weighting schemes (Wang et al., 2021a; 2022b; McClenny & Braga-Neto, 2020; 2023; Maddu et al., 2022) and adaptive resampling of collocation points, such as

importance sampling (Nabian et al., 2021), evolutionary sampling (Daw et al.), and residual-based adaptive sampling (Wu et al., 2023).

Combining traditional methods with DNNs is a new line of research which focuses on leveraging the advantages of both. For example, incorporating a Fourier feature layer to preprocess DNN inputs enhances their capacity to capture high-frequency functions (Tancik et al., 2020), while also reducing eigenvector bias in PINNs (Wang et al., 2021b; Raynaud et al., 2022). Fourier neural operators (Li et al., 2020) utilize Fourier layers in the network architecture and have become a popular approach for inverse problems. Similarly, physics-informed radial basis networks (PIRBNs) have been proposed as efficient local approximators that integrate domain knowledge during training, performing well in solving PDEs with high-frequency features or ill-posed domains (Bai et al., 2023; Fu et al., 2024). Chrysos et al. (2022; 2020) augment polynomial neural networks with DNNs, with a direct focus on classification and discriminative problems. Further explorations encompass alternative objective functions, such as those employing numerical differentiation techniques Huang & Alkhalifah (2024) and variational formulations inspired by Finite Element Methods (FEM) Kharazmi et al. (2019); Berrone et al. (2022); Patel et al. (2022), along with additional regularization terms to expedite PINNs' convergence Yu et al. (2022b); Son et al. (2021).

Recent work has investigated deep architectures for solving PDEs through methods such as stacked DNNs (Howard et al., 2023) and residual blocks with adaptive gating parameters (Wang et al., 2024a). Wang et al. (2024a) incorporate residual blocks with adaptive skip connections, which dynamically balance the input and learned residuals at each layer. Specifically, they incorporate adaptive residual blocks which dynamically balance the contributions of both the input and residual function at each layer, while the stacked approach of Howard et al. (2023) utilizes transfer learning to improves the ability of PINNs to handle complex, multi-scale PDEs.

While these deep residual based approaches show much promise, the increased architectural complexity, leads to higher computational costs both in terms of memory and training time—requiring careful selection of training routines to prevent instabilities or poor convergence. Finally, while Fourier features help in capturing high-frequency components, they can still struggle with discontinuous solutions or sharp interfaces, which are challenging for neural networks to approximate due to the smoothness of typical activation functions.

To address these challenges, we propose a novel class of network architectures termed HyResPINNs, that combines a standard DNN with a Radial Basis Function (RBF) network within bi-level adaptive residual blocks, such that the relative contribution of both the DNN and RBF networks are adaptively learned during training, along with the full residual block outputs. The proposed architecture leverages the strengths of both smooth and non-smooth function approximators to model complex physical systems. The standard DNN using smooth activation functions effectively captures the continuous, global behaviors of the solution, which are common in many physical phenomena. While the RBF network then captures the localized, discontinuous or sharp features in the target solution. This representational division ensures that the model can accurately capture both smooth and non-smooth components of the solution, forming a robust network capable of modeling many types of physical systems.

Our main contributions are summarized as follows:

- **Propose a novel adaptive residual architecture:** We introduce a new class of physics-informed neural networks, HyResPINN, that combine standard neural networks with Radial Basis Function (RBF) networks within bi-level adaptive residual blocks.

- **Demonstrate the superiority of the block structure:** We show that our residual block architecture provides significant improvements over standard approaches in capturing both smooth and non-smooth features, leading to more accurate modeling of complex physical systems.

- **Highlight the benefits of adaptivity between residual blocks:** We demonstrate that the adaptive learning of contributions between the NN and RBF networks, as well as between residual blocks, results in superior performance and stability compared to non-adaptive methods.

- **Thorough empirical evaluation:** Show our method HyResPINN outperforms standard PINNs along with state of the art methods in PINNs on an array of baseline problems, confirming the general applicability of HyResPINNs.

## 2 BACKGROUND

### 2.1 PHYSICS-INFORMED NEURAL NETWORK

Given the spatio-temporal domain $\Omega \subset \mathbb{R}^d$ defined on $[0, T] \times \Omega \subset \mathbb{R}^{1+d}$ where $\Omega$ is a bounded domain in $\mathbb{R}^d$ with regular enough boundary $\partial\Omega$, the general form of a parabolic PDE is,

$$\mathbf{u}_t + \mathcal{F}[\mathbf{u}] = \mathbf{f}, \tag{1}$$

such that $\mathcal{F}[\cdot]$ is a linear or nonlinear differential operator, and $\mathbf{u}(t, \mathbf{x})$ denotes a unknown solution.

The general initial and boundary conditions can be then formulated as:

$$\mathbf{u}(0, \mathbf{x}) = \mathbf{g}(\mathbf{x}), \quad \mathbf{x} \in \Omega, \tag{2}$$
$$\mathcal{B}[\mathbf{u}] = 0, \quad t \in [0, T], \ \mathbf{x} \in \partial\Omega. \tag{3}$$

Here, $\mathbf{f}$ and $\mathbf{g}(\mathbf{x})$ are given functions with certain regularity; $\mathcal{B}[\cdot]$ denotes an abstract boundary operator, representing various boundary conditions such as Dirichlet, Neumann, Robin, and periodic conditions.

We aim at approximating the unknown solution $\mathbf{u}(t, \mathbf{x})$ by a deep neural network $\mathbf{u}_\theta(t, \mathbf{x})$, where $\theta$ denotes the set of all trainable parameters of the network (e.g., weights and biases). If a smooth activation function is employed, $\mathbf{u}_\theta$ provides a smooth representation that can be queried for any $(t, \mathbf{x})$. The PDE residuals are defined as,

$$\mathcal{R}_{\text{int}}[\mathbf{u}_\theta](t, \mathbf{x}) = \frac{\partial \mathbf{u}_\theta}{\partial t}(t, \mathbf{x}) + \mathcal{F}[\mathbf{u}_\theta](t, \mathbf{x}) - \mathbf{f}(\mathbf{x}), \quad (t, \mathbf{x}) \in [0, T] \times \Omega, \tag{4}$$

and spatial and temporal boundary residuals, respectively, by

$$\mathcal{R}_{\text{bc}}[\mathbf{u}_\theta](t, \mathbf{x}) = \mathcal{B}[\mathbf{u}_\theta](t, \mathbf{x}), \quad (t, \mathbf{x}) \in [0, T] \times \partial\Omega, \tag{5}$$

and

$$\mathcal{R}_{\text{ic}}[\mathbf{u}_\theta](\mathbf{x}) = \mathbf{u}_\theta(0, \mathbf{x}) - \mathbf{g}(\mathbf{x}), \quad \mathbf{x} \in \Omega. \tag{6}$$

Then, we train a physics-informed model by minimizing the following composite *empirical loss*:

$$\mathcal{L}(\theta) := \underbrace{\frac{1}{N_{ic}} \sum_{i=1}^{N_{ic}} \left| \mathcal{R}_{\text{ic}}[\mathbf{u}_\theta](\mathbf{x}_{ic}^i) \right|^2}_{\mathcal{L}_{ic}(\theta)} + \underbrace{\frac{1}{N_{bc}} \sum_{i=1}^{N_{bc}} \left| \mathcal{R}_{\text{bc}}[\mathbf{u}_\theta](t_{bc}^i, \mathbf{x}_{bc}^i) \right|^2}_{\mathcal{L}_{bc}(\theta)} + \underbrace{\frac{1}{N_r} \sum_{i=1}^{N_r} \left| \mathcal{R}_{\text{int}}[\mathbf{u}_\theta](t_r^i, \mathbf{x}_r^i) \right|^2}_{\mathcal{L}_r(\theta)},$$
$$\tag{7}$$

which aims to enforce the neural network function $\mathbf{u}_\theta$ to satisfy the PDEs (1) with initial and spatial boundary conditions (2)–(3). The training data points $\{\mathbf{x}_{ic}^i\}_{i=1}^{N_{ic}}$, $\{t_{bc}^i, \mathbf{x}_{bc}^i\}_{i=1}^{N_{bc}}$ and $\{t_r^i, \mathbf{x}_r^i\}_{i=1}^{N_r}$ can be the vertices of a fixed mesh or points randomly sampled at each iteration of a gradient descent algorithm.

### 2.2 RADIAL BASIS FUNCTION NETWORKS

The general form of an RBF network for the training data points $\mathbf{x} = \{\mathbf{x}_i\}_{i=1}^N$ with corresponding function values $y_i$ is given by,

$$f(\mathbf{x}) = \sum_{j=1}^{N_c} c_j \psi(||\mathbf{x} - \mathbf{x}_j^c||), \tag{8}$$

where $\mathbf{x}^c = \{\mathbf{x}_j^c\}_{j=1}^{N_c}$ are the center points of the radial basis functions (RBFs). Here, $\psi$ represents the radial basis function, which are commonly chosen as the Gaussian, multi-quadric, or inverse

multi-quadric functions, each offering different characteristics and benefits. The elements of the kernel matrix are given by:

$$\mathbf{K}_{ij} = \psi(||\mathbf{x}_i - \mathbf{x}_j^c||), \tag{9}$$

where $\mathbf{x}_i$ represent the data points, and (9) evaluates the RBF based on the distance between the points. The kernel matrix captures the pairwise interactions between data points through the chosen RBF, and its structure is typically symmetric and positive-definite, assuming an appropriate choice of $\psi$. The choice of center points can vary depending on the application; in some cases, the center points may coincide with the data points, but they may also be selected independently of the data points to optimize approximation accuracy or computational efficiency. The RBFs determine how much influence each center has on the input, based on the distance between the input and the center.

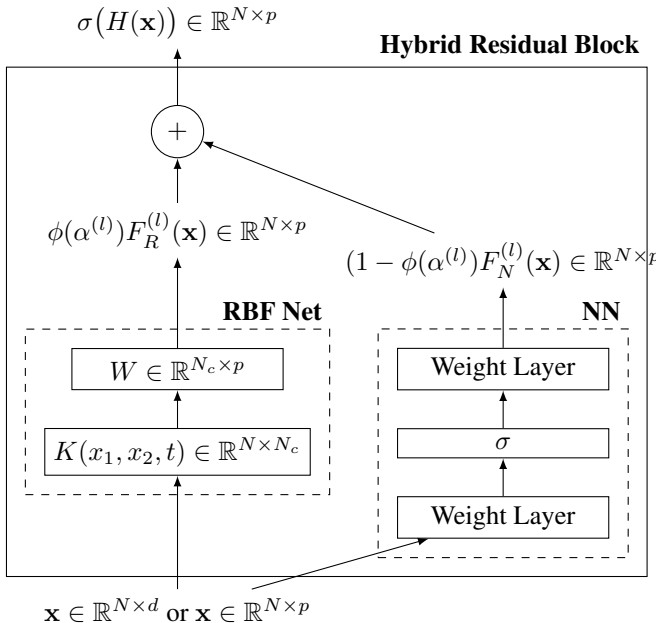

Figure 1: Illustration of the RBF+PINN hybrid residual block with trainable strength connections between the RBF and PINN outputs.

Gradient descent is sometimes used to optimize both the centers and the unknown weights ($c_j$) simultaneously. These approaches are referred to as RBF networks, where the radial basis functions are the hidden layer activations, and the RBF coefficients are the trainable network parameters. The mean-squared error between the linear combination of the hidden layer outputs and the known functions values is commonly minimized.

## 3 HYBRID RESIDUAL PINNS (HYRESPINNS)

In this section, we describe our proposed architecture—HyResPINNs. HyResPINNs are a novel type of residual network such that each residual block incorporates an RBF kernel along with a standard DNN.

### 3.1 HYBRID RESIDUAL BLOCKS

Figure 1 shows the hybrid residual block specifics. We denote the output of the $l$-th residual block for the input $\mathbf{x}^{(l)}$ as $H^{(l)}(\mathbf{x}^{(l)})$. Formally, the forward pass of each hybrid residual block in the

HyResPINN architecture is,

$$H^{(l)}(\mathbf{x}^{(l)}) = \phi(\alpha^{(l)})F_R^{(l)}(\mathbf{x}^{(l)}) + (1 - \phi(\alpha^{(l)}))F_N^{(l)}(\mathbf{x}^{(l)}), \tag{10}$$

such that $\alpha^{(l)} \in \mathbb{R}$ is a trainable parameter, $\phi$ represents the sigmoid function, $F_R^{(l)}(\mathbf{x}^{(l)})$ is the output of the RBF network and $F_N^{(l)}(\mathbf{x}^{(l)})$ is the output from a DNN or PINN within the $l$-th block. We then pass $\sigma(H^{(l)}(\mathbf{x}^{(l)}))$ as the input to the next block, where $\sigma$ is some non-linear activation function.

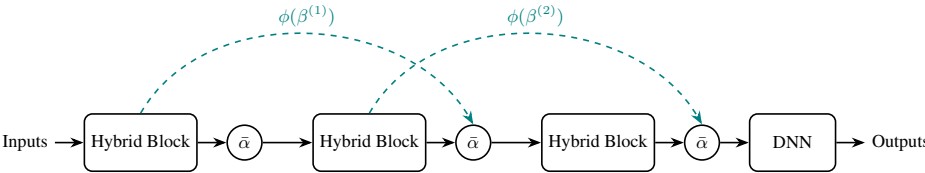

Figure 2: Illustration of the HyResPINN architecture using three blocks.

**Adaptive Hybrid Contribution Coefficients** The trainable parameter $\alpha^{(l)}$ controls the relative contributions of the DNN and RBF components to the output—where optimizing $\alpha$ finds the best balance between these two elements. Since the sigmoid function $\phi$ constrains its output to the range $[0, 1]$, the resulting output is a convex combination of the RBF and DNN components. We further incorporate adaptable residual connection parameters between each hybrid block, denoted as $\beta^{(l)}$—similar to the approaches in (Howard et al., 2023) and (Wang et al., 2024a)

However, unlike (Wang et al., 2024a), where their adaptable parameters are initialized to zero to force the network to learn non-linearities from scratch, we initialize each $\phi(\alpha^{(l)}) = 0.5$, ensuring equal contributions from both components at the start of training, and each $\beta^{(l)} = 1$. When $\phi(\alpha) = 1$, only the RBF network contributes; when $\phi(\alpha) = 0$, only the DNN contributes. The optimal choice of $\alpha$ depends on the problem characteristics. For problems with large regions of smoothness, the model might favor a lower $\alpha$, assigning more weight to the DNN components in each residual block. Conversely, for problems involving sharp transitions or discontinuities, a higher $\alpha$ (favoring the RBF) may be preferable. In hybrid problems with smooth and non-smooth regions, $\alpha$ will likely fall between 0.4 and 0.6, providing a balanced combination of the two networks.

To encourage smoother solutions and prevent the network from introducing excessive non-linearity, we add a regularization term to each block's trainable parameter $\alpha$. This regularization penalizes large values of $\alpha$, effectively controlling the contribution of non-linearity from the RBF components. Specifically, the total loss function is defined as:

$$\mathcal{L} = \lambda_{ic}\mathcal{L}_{ic}(\boldsymbol{\theta}) + \lambda_{bc}\mathcal{L}_{bc}(\boldsymbol{\theta}) + \lambda_r\mathcal{L}_r(\boldsymbol{\theta}) + \lambda_p \sum_{i=1}^{N_{blocks}} \alpha_i^2, \tag{11}$$

where $\mathcal{L}_{ic}$ and $\mathcal{L}_{bc}$ represent the loss terms for the initial and boundary conditions, $\mathcal{L}_r$ represents the residual loss, and the final term $\lambda_p \sum_{i=1}^{N_{blocks}} \alpha_i^2$ acts as an $L_2$ regularization on the $\alpha$ parameters. The regularization helps balance the contributions of the smooth neural network and non-linear RBF components, promoting smoother solutions and more stable training.

**Adaptive RBF Block Kernel** The output of the RBF kernel $F_R^{(l)}(\mathbf{x}^{(l)})$ can be described as,

$$F_R^{(l)}(\mathbf{x}^{(l)}) = K(\mathbf{x})W, \tag{12}$$

such that the kernel matrix $K$ is defined as (9) where the kernel matrix $K$ is based on the Wendland $C^4$ kernel in 2D. In this work, we focus on isotropic version of the Wendland kernel.

The isotropic Wendland $C^4$ kernel for input $\mathbf{x}$ and the $i$-th center is defined as:

$$\psi_i(\mathbf{x}) = \left(1 - \frac{\|\mathbf{x} - \mathbf{x}_i^c\|}{\tau_i}\right)_+^6 \left(35\left(\frac{\|\mathbf{x} - \mathbf{x}_i^c\|}{\tau_i}\right)^2 + 18\frac{\|\mathbf{x} - \mathbf{x}_i^c\|}{\tau_i} + 3\right), \tag{13}$$

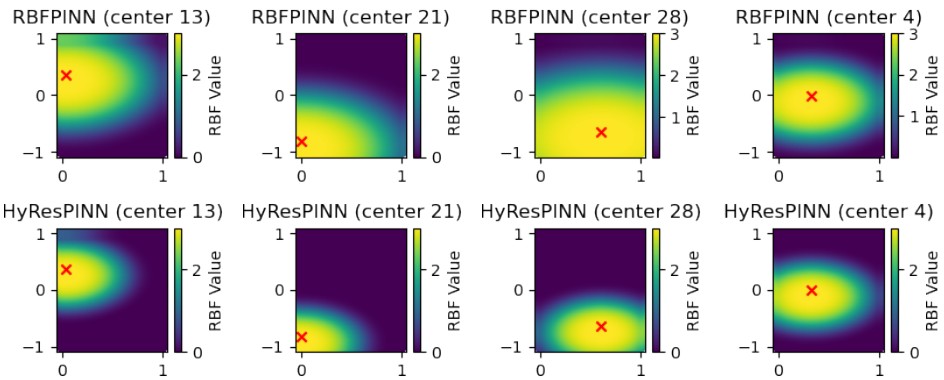

Figure 3: *2D Allen-Cahn equation:* Comparison of (a subset of) the learned RBF kernels for a standard RBF network (top row) and the proposed hybrid RBF+NN residual block approach (bottom row). Each subplot shows the RBF value corresponding to different learned RBF centers in the input domain, marked by red crosses.

where $\mathbf{x}_i^c$ is the center of the $i$-th RBF, $\tau$ is a scaling parameter, and $\|\mathbf{x} - \mathbf{x}_i^c\|$ represents the Euclidean distance between $\mathbf{x}$ and $\mathbf{x}_i^c$. The Wendland kernel is compactly supported, meaning $\psi_i(\mathbf{x}) = 0$ for $\|\mathbf{x} - \mathbf{x}_i^c\| \geq \tau_i$, which leads to sparse kernel matrices and computational efficiency. Each $\tau_i$ is a trainable parameter in the network and optimize through gradient descent along with all other network parameters. Figure 3 demonstrates learned scale parameters within an optimized RBF kernel.

**Block Neural Network** Following the convention of Cyr et al. (2020), we represent the output of the DNN $F_N^{(l)}(\mathbf{x}^{(l)})$ $F_N^{(l)} \in \mathbb{R}^d \to \mathbb{R}$ of width $w$, as a linear combination of adaptive basis functions given by

$$F_N^{(l)}(\mathbf{x}; a, \theta^h) = \sum_{i=1}^{w} a_i \sigma_i(\mathbf{x}; \theta^h),$$

(14)

where each $a_j$ for $j = 1, .., w$ and $\theta^h$ constitute the weights and biases in the last layer and hidden layers respectively, forming the set of all network parameters $\theta$. Then, each $\sigma_j$ are non-linear smooth activation functions such as Tanh acting on the outputs of the hidden layers. We choose to use a standard DNN architecture, but any DNN architecture such as ResNets, would work. The parameters $\theta$ are computed through some iterative optimization technique. In this work, we use variants of gradient descent methods such as ADAM Kingma & Ba (2015) and L-BFGS Liu & Nocedal (1989).

Figure 2 shows a visualization of our full model architecture with adaptive residual block skip connections, along with the input block structure used to lift the inputs to the desired higher dimension, and the output neural network block used to project each block's output down to the output dimension.

| Problem | Domain | Boundary Cond. | PINN | ResPINN | Expert | Stacked | PirateNet | **HyResPINN** |
|---|---|---|---|---|---|---|---|---|
| Allen-Cahn | 1D Space/Time | Periodic | 0.526 | 0.0027 | 0.00386 | 0.00587 | 0.00022 | **$9.62 \times 10^{-5}$** |
| DarcyFlow | 2D Annulus | Neumann | 0.00075 | 0.00046 | 0.0005 | 0.0009 | $8.71 \times 10^{-5}$ | **$5.44 \times 10^{-5}$** |
| (smooth coefficients) | | Dirichlet | 0.0020 | 0.0014 | 0.00012 | 0.0041 | 0.00017 | **$6.0 \times 10^{-5}$** |
| | 3D Annulus | Neumann | 0.0022 | 0.012 | 0.021 | 0.054 | 0.039 | **0.0012** |
| | | Dirichlet | 0.0085 | 0.0061 | 0.0011 | 0.039 | 0.0013 | **0.0011** |
| (rough coefficients) | 2D Box | Neumann | $6.85 \times 10^{-5}$ | $2.69 \times 10^{-5}$ | 0.00011 | 0.00015 | $5.44 \times 10^{-5}$ | **$1.05 \times 10^{-5}$** |

Table 1: Relative $L^2$ test error results for various PDE problems and baseline methods.

## 4 EXPERIMENTAL RESULTS AND DISCUSSION

In this section, we demonstrate the effectiveness of the proposed HyResPINNs architecture across a diverse collection of benchmark problems compared to the leading baseline methods. We show that

HyResPINNs consistently produce lower relative $L^2$ errors between the predicted and ground truth solutions for the same training set size when compared to the applicable baselines trained under the same experimental procedures and hyperparameters. We compare each method on the 1D non-linear hyperbolic Allen-Cahn equation and the Darcy Flow equation in two and three dimensions with both Dirichlet and Neumann boundary conditions. Our main results are summarized in Table 1.

**Baseline Methods.** We compare the proposed HyResPINN against a suite of baseline methods under exactly the same hyper-parameter settings in each example. Baselines include the standard PINN as originally formulated in (Raissi et al., 2019) (PINN), a PINN based on the architectural guidelines proposed in Wang et al. (2023) (ExpertPINNs), PINNs with residual connections (ResPINNs), PirateNets Wang et al. (2024a), and the stacked PINN approach of (Howard et al., 2023) (StackedPINNs). To ensure a fair comparison, we implemented the approaches detailed in (Raissi et al., 2019; Wang et al., 2024a; 2023; Howard et al., 2023) following the architectural details provided in each.

**Experimental Setup.** We follow similar experimental design procedures as those described in Wang et al. (2022a; 2023; 2024a). A full description of the experimental setups are included in Appendix A, and exact hyper-parameter setting are detailed in Table 2. We train all models on Nvidia A100 GPU running Centos 7.2. Code for our methods and all compared baseline approaches is written in libtorch 1.11 (the C++ version of PyTorch) and will be made publicly available upon publication.

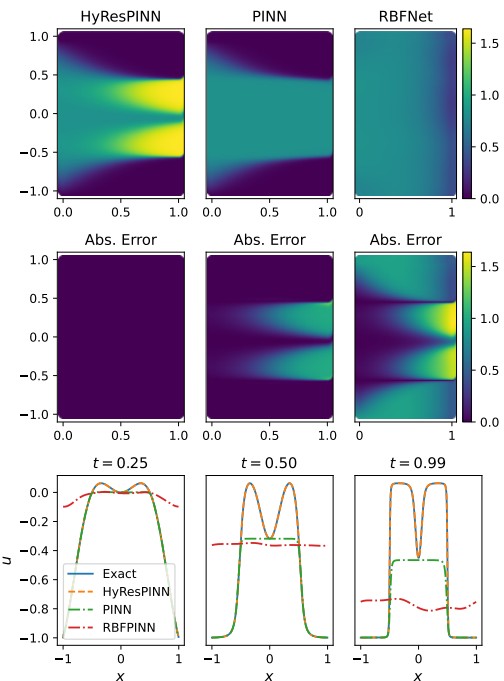

Figure 4: *2D Allen-Cahn equation:* Comparison of the predicted solutions for the Allen-Cahn equation using HyResPINN, standard PINN, and RBF network models. The top row shows the predicted solutions at for HyResPINN (left), standard PINN (center), and RBF network (right). The second row illustrates the absolute error between the predicted solutions and the exact solution. The bottom row shows the predicted solutions for three time steps comparing the exact solution (green), HyResPINN (orange), and standard PINN (dashed blue). The HyResPINN model effectively captures both the smooth and sharp features of the solution, while the standard PINN struggles with sharp transitions, leading to larger errors in these regions.

### 4.1 1D ALLEN-CAHN

We first focus on the Allen-Cahn equation, a challenging benchmark for conventional PINN models that has been extensively studied in recent literature (Wight & Zhao, 2020; Wang et al., 2022a; Daw et al.). For simplicity, we consider the one-dimensional case with a periodic boundary condition with $t \in [0, 1]$, and $x \in [-1, 1]$:

$$u_t - 0.0001u_{xx} + 5u^3 - 5u = 0 \,,$$

$$u(0, x) = x^2 \cos(\pi x) \,,$$

$$u(t, -1) = u(t, 1) \,, \ u_x(t, -1) = u_x(t, 1) \,.$$

The Allen-Cahn PDE is an interesting benchmark for PINNs as it introduces periodic boundary conditions, and because it is a "stiff" PDE that challenges PINNs to approximate solutions with sharp space and time transitions.

**Hybrid residual blocks improve prediction accuracy.** We first demonstrate that the hybrid residual block structure introduced in our method enhances the ability to capture both smooth and sharp features in the 2D Allen-Cahn solution. As shown in Figure 4, the proposed approach outperforms competing baselines, which struggle to accurately represent the sharp transitions in the solution. This difficulty arises from the smoothness constraints of standard neural network architectures. In contrast, our method effectively captures these sharp transitions by leveraging smooth neural networks and adaptive non-smooth RBF kernels, allowing the model to balance global smooth behavior and

localized discontinuities. Figure 3 shows a random subset of the learned RBF kernels captured by the proposed hybrid approach. Specifically, it captures similar RBF structures (to a standard RBF network). However, it benefits from the additional flexibility the NN component provides, allowing for more nuanced function approximation in regions requiring a balance of smoothness and sharp transitions. This is likely apparent in the smaller kernels learned by the hybrid block.

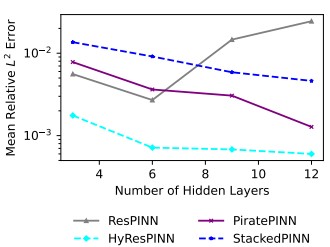

**Stacking structure of residual blocks improve prediction accuracy.** We next show that the stacking structure employed by the HyResPINN in each hybrid residual block structure introduced here further helps to capture the smooth and sharp features in the Allen-Cahn solution. Figure 5 shows the predicted solutions between the proposed method and the competing baselines. The competing methods need help accurately capture the sharp transitions in the solutions. In contrast, our approach more accurately captures these sharp transitions due to the conjunction of the smooth neural network architecture and the learned non-smooth RBF kernels.

Figure 5: *Allen-Cahn equation:* Comparison of the mean relative $L^2$ error using various methods as a function of the number of hidden layers.

Next, we evaluate the effect of network depth on predictive accuracy for different architectures. Figure 5 HyResPINN consistently achieves the lowest error across different network depths. The performance of other methods shows varying sensitivity to the number of hidden layers, with HyResPINN offering the most robust performance. Further, the left plot in Figure 6 shows that HyResPINN and PiratePINN achieve the lowest error over iterations, with HyResPINN consistently reducing the error more effectively than the other methods. The right plot demonstrates that—although HyResPINN has a slightly higher training cost—it significantly outperforms other methods in accuracy, particularly for long training times. These results highlight the robustness and efficiency of HyResPINN in solving complex PDEs.

**Remark** While attempting to replicate the results of (Wang et al., 2024a), using the reported hyperparameters and implementation details, we observed a slight performance gap between the results obtained in our implementation and those listed in their paper. The observed discrepancy in the final model's performance could

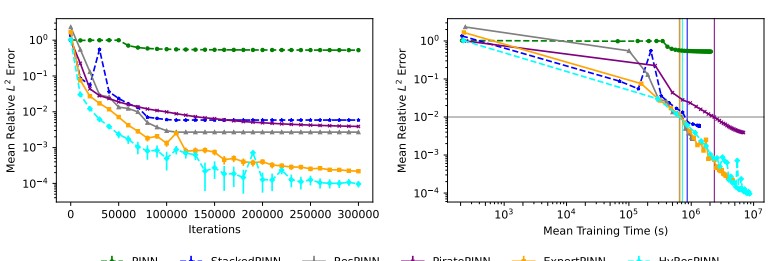

Figure 6: *Allen-Cahn equation:* Comparison of mean relative $L^2$ error across various methods, plotted against the number of training iterations (left) and the mean training time (right). Vertical bars in right plot indicate the time for which each method achieved an error of $10^{-2}$.

stem from several factors such as subtle variations in the implementation environment—such as hardware differences, software library versions, or random seed initialization. Despite the differences between our replication results and those reported in (Wang et al., 2024a), the comparison between our architecture and the replicated version of their approach remains fair, as both were tested under identical conditions using the same dataset, hyperparameters, and training setup (including hardware and software environment). Our findings indicate that our architecture outperforms the replicated version of their model in many cases, along with all other replicated baseline methods.

### 4.2 DARCY FLOW

In this section, we describe the Darcy Flow problem, an elliptic boundary value problem given by,

$$-\nabla \cdot \mu \nabla \phi = f \quad \text{in } \Omega \tag{15}$$

$$\phi = u \quad \text{on } \Gamma_D$$

$$\mathbf{n} \cdot \mu \nabla \phi = g \quad \text{on } \Gamma_N$$

where $\Gamma_D$ and $\Gamma_N$ denote Dirichlet and Neumann parts of the boundary $\Gamma$, respectively, $\mu$ is a symmetric positive definite tensor describing a material property and $f$, $u$ and $g$ are given data.

In important applications such as porous media flow, heat transfer, and semiconductor devices, the flux $\mathbf{u} = -\mu \nabla \phi$ is the variable of primary interest. Specifically, during the fabrication of semiconductor devices, impurities are introduced into the silicon substrate to alter its electrical properties—a process known as doping. The diffusion of dopants can be described by equations that are similar in form to Darcy's law, where the dopant concentration gradient drives the diffusion process. In semiconductor doping, impurities are introduced into the substrate to change its electrical properties. This involves diffusion, where the concentration gradient drives the movement of dopant atoms into the semiconductor material. While Darcy's law governs fluid flow through porous media based on pressure gradients, the diffusion of dopants in semiconductors is guided by concentration gradients, analogous to how fluids move through porous materials in Darcy's law.

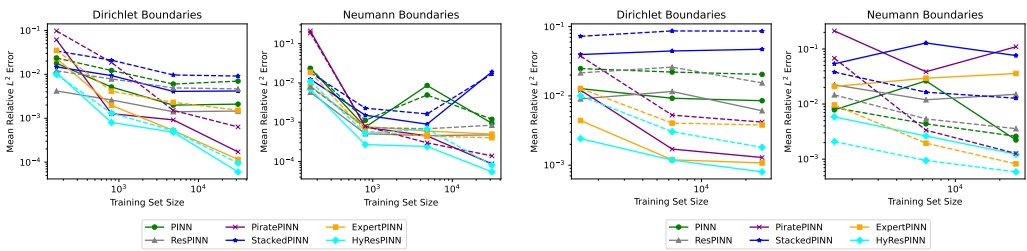

Figure 7: *2D and 3D Smooth Darcy Flow equation:* Comparison of the mean relative $L^2$ errors across each baseline method for the 2D Darcy Flow problem (left) and the 3D Darcy flow problem (right) plotted against the number of training collocation points. Solid lines show the solution error, while dashed lines show the x-directional flux errors.

### 4.2.1 ANNULUS DOMAIN WITH SMOOTH COEFFICIENTS

We initially showcase the convergence rate of our method by employing smooth manufactured solutions. We consider two scenarios: one within an annular 2D domain, and another within a 3D domain formed by extruding the 2D geometry in the z-direction, resulting in a cylinder with a height of two. We present a convergence study analyzing the error rate as the number of points increases. The exact solutions are set as,

$$u(x, y) = \sin(x) \sin(y), \quad u(x, y, z) = \sin(x) \sin(y) \sin(z), \tag{16}$$

for the 2D and 3D cases, respectively. By substituting these exact solutions into the model problem we define the source term and the boundary data. We solve the discrete problem with either Dirichlet or Neumann boundary conditions. Convergence results are shown in Figure 7 for the two and three dimensional problems and demonstrate that HyResPINNs remain robust to boundary condition type, size of training set (achieves lowest errors compared to the baselines for fewer training points), and problem dimension.

### 4.2.2 2D BOX DOMAIN WITH ROUGH COEFFICIENTS

A defining feature of methods for solving the Darcy Flow equation 4.2, lies in their proficiency in accurately depicting the flux variable in scenarios where the coefficients $\mu$ are discontinuous. In these instances, the flux's normal component maintains continuity across material interfaces, whereas the tangential component may exhibit discontinuities. In contrast, collocated methods like Galerkin, stabilized Galerkin, and least-squares finite elements often fail to replicate this physical behavior, typically resulting in oscillations at the interface. In this section, we demonstrate that HyResPINNs successfully provides physically accurate flux approximations for problems characterized by discontinuous coefficients.

The initial example presented is the well-documented "five strip problem" Nakshatrala et al. (2006); Masud & Hughes (2002), which serves as a conventional manufactured solution test to evaluate a method's capability to preserve the continuity of normal flux. The prescribed exact solution on domain $\Omega = [0, 1]^2$ is given by,

$$\phi_{ex} = 1 - x, \quad \text{and} \quad \Gamma_N = \Gamma, \tag{17}$$

such that $\Omega$ is divided in five equal strips,

$$\Omega_i = \{(x, y) \mid 0.2(i - 1) \ \leq \ y \ \leq \ 0.2i \ ; \ 0 \leq x \leq 1\}, \quad i = 1, ..., 5 \tag{18}$$

with different $\mu_i$ on each $\Omega_i$ such that $\mu_1 = 16$, $\mu_2 = 6$, $\mu_3 = 1$, $\mu_4 = 10$, $\mu_5 = 2$. We solve the discrete problem with Neumann boundary conditions. Convergence results are shown in Figure 8 and show that HyResPINNs provide accurate solutions, albeit with larger training sizes.

## 5 CONCLUSION

In this work, we introduced HyResPINNs, a novel class of physics-informed neural networks that incorporate adaptive hybrid residual blocks combining the strengths of standard neural networks and radial basis function (RBF) networks. Our architecture effectively captures continuous and discontinuous features by leveraging smooth and non-smooth function approximators. The adaptive combination parameters within each block allow the model to balance the contributions of neural and RBF components during training. Furthermore, including Wendland kernels enhances the model's ability to handle sharp transitions while maintaining computational efficiency. Our experiments demonstrate that HyResPINNs outperform traditional PINNs and state-of-the-art methods in accuracy and robustness, particularly for problems involving mixed smooth and non-smooth regions. This work increases the flexibility and generalizability of PINNs by bridging the gap between classical numerical methods and DNN-based approaches for solving PDEs.

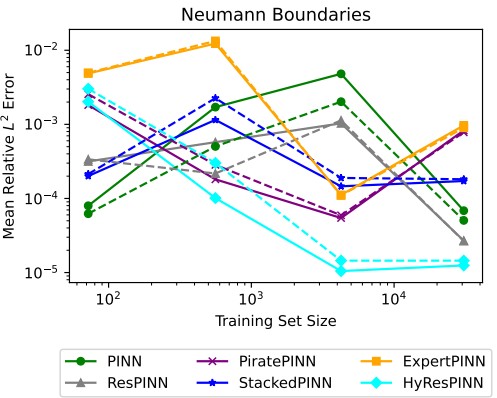

Figure 8: *2D Rough Darcy Flow equation:* Comparison of the mean relative $L^2$ errors across various methods, plotted against the number of training collocation points. Solid lines show the solution error, while dashed lines show the x-directional flux errors.

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

## A EXPERIMENTAL DESIGN

We use mini-batch gradient descent, where collocation points are randomly sampled during training iteration, for all Allen-Cahn experiments, and we use full-batch gradient descent (but varying training set size) for all other experiments. We use the Adam optimizer Kingma & Ba (2015), and follow the learning rate schedule of Wang et al. (2023) which starts with a linear warm-up phase of $5,000$ iterations, starting from zero and gradually increasing to $10^{-3}$, followed by an exponential decay at a rate of $0.9$. Following the best practices described in Wang et al. (2023), we also employ a learning rate annealing algorithm Wang et al. (2023) to balance losses and causal training Wang et al. (2022a; 2023) to mitigate causality violation in solving time-dependent PDEs and apply exact periodic boundary conditions Dong & Ni (2021) when applicable. We use the hyperbolic tangent activation functions and initialize each network's parameters using the Glorot normal scheme, unless otherwise specified. We ran five random trials for each test, and report the mean values achieved in each plot and table.

| Parameter | Value |
|---|---|
| **Architecture** | |
| Number of layers | 9 |
| Number of channels | 128 |
| Activation | Tanh |
| Fourier feature scale | 2.0 |
| Random weight factorization | $\mu = 1.0, \sigma = 0.1$ |
| **Learning rate schedule** | |
| Initial learning rate | $10^{-3}$ |
| Decay rate | 0.9 |
| Decay steps | $5 \times 10^3$ |
| Warmup steps | $5 \times 10^3$ |
| **Training** | |
| Training steps | $3 \times 10^5$ |
| **Weighting** | |
| Weighting scheme | Gradient Norm Wang et al. (2022b; 2023) |
| Causal tolerance | 1.0 |
| Number of chunks | 32 |

Table 2: Hyper-parameter configurations for experiments.

