# OpenReview forum: "HyResPINNs: Adaptive Hybrid Residual Networks for Learning Optimal Combinations of Neural and RBF Components for Physics-Informed Modeling"
_ICLR.cc/2025/Conference — Submitted to ICLR 2025_

### Official Review · Reviewer_6pTT · 2024-11-02

**Soundness:** 3
**Presentation:** 3
**Contribution:** 2
**Rating:** 5
**Confidence:** 4

**Summary:**

The work proposes an architecture with the adaptive residual connection between a regular neural network and a Radial bias network. They also demonstrated the effect of the residual connection and adaptivity of the residual connection. The designed architecture has been shown to outperform baselines on Allen-Cahn and Darcy flow.

**Strengths:**

The authors tackle a significant issue in scientific computation. The architecture they propose is clearly explained in the paper. The authors conducted ablation studies to illustrate the importance of the proposed components. The proposed model demonstrates superior performance compared to baseline models.

**Weaknesses:**

1. The motivation of the work was “While these deep residual-based approaches show much promise, the increased architectural complexity leads to higher computational costs both in terms of memory and training time—requiring careful selection of training routines to prevent instabilities or poor convergence” - however, authors do not report computation cost and memory requirement for the baselines. Also, from Figure 6, we notice that PirateNet and the proposed model are rather close when compared to training time.

2. To prove the model's superiority, other PDEs, such as the Navier–Stokes equation, the Grey-Scott equation, the Ginzburg-Landau equation, the Korteweg–De Vries equation, etc., should be used as the baseline.

3. The work replaces the residual connection by the RBF network in PiretNet. This limits the technical novelty of the work

**Questions:**

1. $\alpha$ in Eq. 10 does not depend on the input, right? Will it be useful if $\alpha$ also depends on the input?

2. Line 217: both sigmoid and RBF functions are denoted by $\phi$. Authors should consider renaming to avoid confusion

3. why the RBF kernel is chosen to be the Wendland C4 kernel? At this point, it seems that the choice is arbitrary.

---

> ### Author Response · Authors · 2024-12-04
> **Response to Reviewer 6pTT (part 1)**
>
> We thank the reviewer for their comments. Below, we address each question and comment individually.
>
> > **Weakness # 1**
>
> Thank you for this observation. We appreciate the need to explicitly report computational cost for the baselines, as these metrics are crucial for understanding the practical trade-offs of different approaches. To address this concern, please see our response above to Reviewer ba3x. Regarding the similarity between PirateNet and HyResPINN in terms of training time (as shown in Figure 6), we agree that the differences are less pronounced during certain training regimes. However, we note that our model consistently achieves superior accuracy for a given computational budget, as demonstrated by the faster convergence in mean relative L2 error.
>
> > **Weakness # 2**
>
> We appreciate the reviewer’s comments suggesting the inclusion of additional PDE examples. While the selected problems (e.g., Allen-Cahn and Darcy Flow) were chosen to evaluate distinct aspects of our method, such as handling nonlinear dynamics and varying boundary conditions, we recognize that additional examples would further strengthen the evaluation. If accepted, we plan to expand our experiments to include additional PDEs.
>
> > **Weakness # 3**
>
> We appreciate the reviewer's concern regarding the technical novelty of our approach. While it is true that our work builds on the residual architecture used in PirateNet, the integration of RBF networks is not simply a substitution but rather a concrete enhancement that introduces new capabilities and performance improvements.
>
> Specifically, by leveraging RBF networks, our method achieves a more refined representation of sharp transitions in solutions, as illustrated in Figure 4. RBF kernels' localized and adaptive nature enables the hybrid architecture to balance smooth and sharp features, a capability that PirateNets and the other baselines struggle to achieve. This improvement is further validated by the consistently lower errors across a wide range of PDEs, as demonstrated in Table 1. Moreover, using RBF networks within the residual framework enriches the expressive power of the model. Unlike traditional neural networks, RBF kernels provide localized basis functions that adaptively capture fine-grained solution structures. This integration allows for a more flexible and efficient representation of PDE solutions, particularly for problems with complex features.
>
> While existing architectures inspire our approach, combining neural networks with RBF-based methods for PDE solving represents a novel contribution. This hybridization bridges two previously distinct paradigms and opens new avenues for extending residual-based architectures.
>
> > **Regarding Question # 1**
>
> We appreciate the reviewer’s question regarding whether \( \alpha^{(l)} \) should depend on the input \( \mathbf{x} \). In our current implementation, \( \alpha^{(l)} \) is a trainable scalar shared across the domain. This design choice simplifies the model and reduces the number of parameters while still capturing a wide range of solutions, as shown in our experiments.
>
> That said, making \( \alpha^{(l)} \) dependent on \( \mathbf{x} \) could offer additional flexibility, enabling the model to adapt the contributions of the RBF and neural network components based on local solution features. For example, in regions with sharp transitions, the RBF component might dominate, while in smoother regions, the neural network could take priority. Exploring this idea is a promising direction for future work, and we thank the reviewer for raising it.
>
> > **Regarding Question # 2**
>
> We thank the reviewer for highlighting this notation issue. We have renamed the kernel description in the revised manuscript to distinguish it from the sigmoid function.

---

> ### Author Response · Authors · 2024-12-04
> **Response to Reviewer 6pTT (part 2)**
>
> > **Regarding Question # 3**
>
> We thank the reviewer for raising this question. The Wendland \( C^4 \) kernel was chosen due to its compact support, smoothness, and computational efficiency, making it well-suited for the hybrid architecture. Compactly supported kernels, like Wendland \( C^4 \), result in sparse kernel matrices, which improve scalability and reduce computational overhead, especially for high-dimensional problems. Specifically, the kernel value is zero for points outside a specified radius, $\tau$ around the kernel center. This compact support property means that most entries in the kernel matrix are zero, as interactions only occur between points within the support radius. Consequently, the resulting sparse kernel matrices require less memory for storage and enable the use of efficient sparse matrix solvers, significantly reducing the computational complexity compared to dense kernel methods. Additionally, the \( C^4 \) smoothness ensures sufficient differentiability for solving PDEs that require higher-order derivatives.
>
> While the Wendland \( C^4 \) kernel provides a balanced trade-off between computational efficiency and smoothness, we acknowledge that this choice is one of several possibilities. Other kernel functions, such as Gaussian or Matérn kernels, could also be explored, depending on the problem requirements. For instance, Gaussian kernels are widely used for their universal approximation properties, but they lack compact support, leading to dense kernel matrices that scale poorly with the problem size. Investigating the performance of alternative kernel functions tailored to specific PDEs or problem domains is a promising direction for future work. We appreciate the reviewer’s suggestion to clarify this point, and we will emphasize this discussion in the revised manuscript to better highlight the rationale for our choice and its implications.

---

### Official Review · Reviewer_AtLi · 2024-11-02

**Soundness:** 2
**Presentation:** 4
**Contribution:** 3
**Rating:** 5
**Confidence:** 5

**Summary:**

This paper proposes a novel architecture for Physics Informed Neural Networks (PINNs), combining elements from both deep learning and kernel modeling. Specifically, their architecture, named HyResPINNs implement blocks where a dense layer and a RBF kernel regression are computed in parallel, then combined by an adjustable weighted average. The network also implements trainable residual connections between blocks to help train deeper architectures.

After providing a thorough literature review, the authors formally describe their method, highlighting how the mixed Neural Network/RBF components help the network learn combinations of smooth and high-frequency components of the target function, leveraging the advantages of each method. Their training method also includes a regularization term, penalizing higher contributions of the RBF components in order to limit overfitting to very high-frequency functions. Finally, the authors evaluate their method against several competitive baselines on two PDEs: the Allen-Cahn equation, and the Darcy Flow. Their experiments show HyResPINNs consistently outperforming other methods on these benchmarks.

The authors conclude by summarizing their work, highlighting how their method is able to capture sharp solutions at a manageable computational cost.

**Strengths:**

### Originality
To the best of my knowledge, this work appears to present novel and interesting ideas. By using a very local RBF kernel in order to assist (but not completely replace) traditional neural network solvers for PINNs, their work aims to help address the well-known spectral bias problem for learning highly oscillatory functions.


### Quality
The paper presents a very interesting idea, and details the HyResPINN architecture in an effective manner. Their experiments are grounded and well-executed, and they compare their method against several competitive baselines. The authors also do a good job of summarizing existing work and highlighting the differences between different methods. I also like their idea of regularizing the weight of the RBF contribution of each block to stabilize the training process.


### Clarity
The paper is overall very well written and clear in its explanations. The literature review at the introduction is thorough, and the required information is presented in a clear and concise manner. The figures are informative and capture the author's arguments well. I particularly like figure 3, where they show kernels learned in HyResPINN are more local than ones from RBFPINNs.

**Weaknesses:**

### Limited Experiments
In my view, the biggest setback of this paper is the limited range of PDEs considered in the experiments section. The authors do a good job of executing the experiments included in the paper, but only consider the Allen-Cahn equation and the Darcy Flow problem (under different conditions). It is great to see their method work well in these cases, but I believe the paper would greatly benefit from additional experiments using other PDEs, specially ones that challenge existing PINNs architectures. Examples of PDEs that could be considered, in order of difficulty, include: 1) the Poisson equation with different forcing functions (depending on the forcing function this can become a harder problem); 2) Burger's equation, 3) the advection equation; 4) the Kuramoto–Sivashinsky equation, 5) problems using the Navier-Stokes equation, such as lid-driven cavity flow (even with low Reynolds number). Although not all of these PDEs need to be considered, including at least one or two of them (or other suitable problems) could make for a stronger paper.

### Flawed Notion of Training Set Size in PINNs
Another critique I have is on the experiments/plots where they examine the performance of different architectures using different training set sizes (e.g.: figures 7 and 8). Under the Physics Informed framework, although the target function is in principle unknown, we can query the differential operator $\mathcal{F}$ from equation (1) on any point of the input domain using automatic differentiation. This means that it is possible to sample collocation points at will at any given point in the domain, as the authors mention themselves in line 147. In fact, it is always recommended to sample points randomly and independently across the entire input space at each iteration of the training algorithm, effectively meaning that there is unlimited "training data" available for PINN problems. Not only is this approach (independent random collocation points at each iteration) more theoretically grounded by taking advantage of the mesh-less nature of PINNs, it often leads to more accurate and robust performance. This renders the comparison of different "training sizes" meaningless, as it is always possible (and encouraged) to sample new points.


### Other comments/typos that did not affect my score:
- [line 140] In equation (4) the $\mathcal{D}$ should be $\mathcal{F}$ instead, in order to be consistent with equation (1).
- [Figure 1] There is a typo in the diagram. According to the formula from equation (10) and regularization shown in equation (11), it should be $\phi(\alpha^{(l)})$ multiplying $F_R^{(l)}(x)$ and $(1-\phi(\alpha^{(l)}))$ multiplying $F_N^{(l)}(x)$, not the other way around, as it is currently shown.
- [line 215] In equation (10), the activation $\sigma$ is shown to be part of the function $H^{(l)}$, while in the diagram of Figure 1 the $\sigma$ is shown outside of the function $H$. One of these should be changed for the sake of consistency.
- [line 303] There is a mention of an "input block" that lifts the original input to a higher dimension in the diagram, but Figure 2 does not include this block, only the "output block".
- [line 362] Given the initial conditions, the boundary conditions are not satisfied at $t=0$, making this problem ill-posed as it is (you can check that $u_x(0,-1)=2$, while $u_x(0,1)=-2$). This is, unfortunately, a very common mistake in the PINNs community, as this specific benchmark has now become standard. Instead, a very similar solution is given by assuming zero Dirichlet boundary conditions on $x=\pm1$, which makes the problem well-posed and has also been studied in a couple papers. It likely won't make much of a difference in the results, but if possible I would encourage the authors to run this benchmark with Dirichlet boundary, or at the very least add a remark/footnote about the ill-posed nature of the problem.
- [lines 710 + 712] There seems to be a bad reference pointer in the LaTeX file.

**Questions:**

I list below my questions/suggestions to the authors, in order of how influential they would be towards increasing my score of their submission.

- [**Including More PDEs In Experiments**] As mentioned above, I believe including more experiments with other PDEs would make for a stronger paper. Suggestions for which PDEs to use are detailed in the previous section. I would be open to increasing my score if more experiments are conducted, even if HyResPINNs don't necessarily beat all other baselines on them.

- [**Reconsidering Notion of "training size" For PINNs + Randomly Sampling Collocation Points**] As mentioned in the previous section, I would urge the authors to move away from the notion of "training size" for PINNs, and train networks with freshly-sampled random collocation points at each iteration.

- [**Clarity on Formulation of Residual Connections**] The adaptive residual connections are implicitly defined in line 234, but it would be good do add an extra formula detailing it. Are the $\beta^{(l)}$ parameters constrained in any way, or is it left for the sigmoid function $\phi$ to make the residual connection in the $(0,1)$ interval? If so, initializing the $\beta^{(l)}$ to be 1, as indicated in line 237, leads the residual connection to have strength $\phi(1) \approx 0.73$. Is there a particular meaning over this choice? In a related issue, initializing the $\alpha^{(l)}$ to be 0.5 means that $\phi(0.5)\approx 0.62$, which is not an equal contribution, as mentioned in line 237. Did you mean to say the $\alpha^{(l)}$ are initialized to be 0?

- [**Other RBF Kernel Possibilities**] The choice of using the Wendland kernel seems well motivated to me, and overall a good choice, but it would be great to see the effect of using other kernels in the RBF blocks of HyResPINNs. This could be a valuable ablation to include, either in the main text, or the appendix. Such an ablation could be done for a single PDE, if testing for all problems is too troublesome/time-consuming.

- [**Reporting Computation Time + Hardware**] The authors highlight that HyResPINNs offer significant gains over plain MLPs, with little extra computational cost, but the training time for each method is not reported, to the best of my knowledge. It would also be good to specify the hardware used to run the experiments.

- [**Lack of Code As Supplementary Material**] In order to get a better understanding of the method and evaluate the execution of experiments, it would be good to provide representative code for running experiments as part of the supplementary material, which is currently missing.

- [**References for RBF Networks**] Overall, the authors do a good job at highlighting existing work and providing a careful perspective of current PINNs research. However, there are no references given for RBF networks, either in the introduction or in section 2.2.

---

> ### Comment · Reviewer_AtLi · 2024-12-02
> **End Of Discussion Period Soon**
>
> Hello,
>
> I just wanted to confirm the authors have not made any changes/replies to their submission, is that correct? As the discussion period is set to end today, there might not be enough time for a timely response from reviewers, should the authors decide to reply to our reviews.

---

> ### Author Response · Authors · 2024-12-04
> **Response to Reviewer AtLi (part 1)**
>
> We thank the reviewer for their comments. Below, we address each question individually.
>
> > **Including More PDEs In Experiments**
>
> We appreciate the reviewer’s comments suggesting the inclusion of additional PDE examples. While the selected problems (e.g., Allen-Cahn and Darcy Flow) were chosen to evaluate distinct aspects of our method, such as handling nonlinear dynamics and varying boundary conditions, we recognize that additional examples would further strengthen the evaluation. If accepted, we plan to expand our experiments to include additional PDEs.
>
> > **Reconsidering Notion of "training size"**
>
> We thank the reviewer for highlighting this aspect of PINN training and for pointing out the advantages of sampling independent random collocation points at each iteration. We agree that this approach often leads to more accurate and robust performance, and we have employed this strategy for the Allen-Cahn experiments, as detailed in the Appendix. Specifically, we mention that collocation points were randomly sampled during each training iteration for this experiment, utilizing the mesh-free nature of PINNs.
>
> For other experiments, such as Darcy flow, we used fixed collocation points instead, as this setup is more commonly employed for comparison against baseline methods in the existing literature. This choice also allowed us to explore the impact of training set size on the performance of various architectures, which remains a common experimental protocol in the PINN community. While we recognize that this approach may not leverage the full flexibility of PINNs, it provides meaningful comparisons in contexts where fixed-point setups are standard.
>
> > **Clarity on Formulation of Residual Connections**
>
> We thank the reviewer for their feedback and for pointing out areas where further clarity is needed regarding the formulation of the residual connections and initialization of parameters. In our current implementation, \( \beta^l \) is initialized to 10, ensuring that \( \phi(\beta^l) \approx 1 \). Similarly, \( \alpha^l \) is initialized to 0, such that \( \phi(\alpha^l) = 0.5 \), ensuring an equal balance between the contributions of the RBF and NN components within each block at the start of training. While the \( \alpha^l \) values are regularized during training, the \( \beta^l \) parameters are constrained only by the sigmoid function \( \phi \), ensuring their values remain within \( (0, 1) \) for stable training. By initializing \( \phi(\beta^l) \approx 1 \), the residual connections initially fully pass the outputs from previous blocks. Similarly, the equal weighting of \( \phi(\alpha^l) = 0.5 \) ensures neither component dominates prematurely. We will add these clarifications to the revised manuscript and further discuss the roles of \( \beta^l \) and \( \alpha^l \) and how their adaptive modulation contributes to the hybrid model's flexibility and performance.
>
> > **Other RBF Kernels**
>
> We thank the reviewer for suggesting exploring alternative kernels in the RBF blocks of HyResPINNs. We agree that investigating the effect of other kernels, such as Gaussian or Matérn kernels, could provide valuable insights into the flexibility and robustness of the hybrid architecture. If the paper is accepted, we will include an ablation study in the appendix, evaluating the performance of HyResPINNs with different kernels on a representative PDE problem. This experiment will allow us to assess the trade-offs between kernel choices regarding accuracy, efficiency, and adaptability to varying solution characteristics. We believe this addition will enrich the paper and further clarify the impact of the kernel selection on the proposed framework.
>
> > **Reporting Computation Time**
>
> We appreciate the reviewer’s suggestion to provide more details on computational time and the hardware used for experiments. In the manuscript, we mentioned the training hardware (NVIDIA A100 GPU running CentOS 7.2) in the "Experimental Setup" section. Additionally, we included a brief computational cost analysis in Figure 6 (see our response to reviewer ba3x). However, we acknowledge that the explicit reporting of training times for each method and experiment was not fully detailed. To address this, we have now included the training time for each method in the table above (see our response to reviewer ba3x). This table compares wall-clock training times for all methods.

---

> ### Author Response · Authors · 2024-12-04
> **Response to Reviewer AtLi (part 2)**
>
> > **Code as supplementary material**
>
> We fully agree that providing code is essential for reproducibility and for gaining a better understanding of the method. We plan to release the complete codebase for our methods upon acceptance of the paper. This code will include scripts for training and evaluating the proposed HyResPINN architecture and reproducing all experimental results presented in the manuscript. The release will ensure that readers and researchers can evaluate the execution of experiments and extend the work easily. We have added a note in the "Experimental Setup" section clarifying our commitment to making the code publicly available upon publication.
>
> > **RBF Netowork References**
>
> We thank the reviewer for pointing out the lack of references for Radial Basis Function (RBF) networks in the Introduction and Section 2.2. We agree that including relevant references would strengthen the context and situate our work more effectively within the broader literature. To address this, we have added key references to RBF networks in the introduction (see Line 062), including (Bai et al. 2023) and (Fu et al. 2024), who use physics-informed RBF networks to solve PDEs.
>
>
> Finally, we thank the reviewers for their careful reading of our manuscript and for pointing out the grammatical, typographical, and notational errors. In the revised manuscript, we have updated the notation in Equations (1) and (10), fixed the typo in the diagram, and fixed the reference pointers in the Appendix. We will additionally add a footnote acknowledging the ill-posed nature of the problem and referencing relevant papers that address it with zero Dirichlet boundary conditions.

---

### Official Review · Reviewer_ba3x · 2024-11-03

**Soundness:** 3
**Presentation:** 3
**Contribution:** 3
**Rating:** 5
**Confidence:** 5

**Summary:**

This contribution proposes a method based on residual blocks of hybrid RBF networks and standard MLPs. The method follows the general approach of PirateNets and Stacked networks in that residual blocks are used, but here it appears that there are no gating mechanisms to allow the model to start from a small configuration and progressively add more blocks during training, as in PirateNets.

**Strengths:**

The method is part of a recent trend in using stacked networks/multifidelity approaches, which appear to improve the accuracy of PINNs.

The contribution of the RBF-NN and MLP is adaptively learned in each block.

The paper is generally well written, with the presence of a small number of typos.

**Weaknesses:**

There is no discussion of the computational complexity of the proposed approach. Nothing is said about training and inference time in comparison with baseline approaches.

Only two benchmark PDEs are used in the experiments, which is not enough to evaluate the effectiveness of the proposed approach.

It is claimed that the RBF-NN can improve the approximation of sharp transitions in the solution, but no detailed plots or discussion is given in support of that. In particular,  Figure 3 seems to contradict this claim, as the kernels look very smooth.

**Questions:**

Are the centers and coefficients trained in each RBF-NN?

A few typos need to be corrected:

Eq (4), script D should be script F.

Bottom of page 7: repetition of "smaller".

Appendix A has broken references.

---

> ### Author Response · Authors · 2024-12-04
> **Response to Reviewer ba3x (part 1)**
>
> We thank the reviewer for their comments. Below we address each weakness individually.
>
> > **Regarding the computational complexity of the proposed approach.**
>
> Thank you for highlighting the importance of discussing computational complexity. In the original submission, we included an analysis of computational cost in Figure 6, where the right plot explicitly compares the mean wall-clock training time and error across different methods, including our proposed approach (HyResPINNs). As shown, while HyResPINN has a slightly higher (overall) training cost, it outperforms baseline methods in terms of accuracy. Specifically, the vertical lines in the figure indicate how long each model trained until achieving a relative L2 error of 10^-2. The figure shows that ResPINNs, ExpertPINNs, and HyResPINNs (ours) achieve this threshold at similar wall-clock times, while PirateNets took the longest. This trade-off between training time and accuracy highlights the robustness and efficiency of our approach to solving complex PDEs.
>
> However, to further address your concerns, the table below includes a more detailed breakdown of computational complexity. Specifically, this table analyzes the wall-clock training time (in minutes) of HyResPINNs compared to each baseline approach corresponding to the best accuracy results presented in Table 1.
>
> | Problem           | Domain        | Boundary Cond. | PINN   | ResPINN | Expert  | Stacked  | PirateNet | **Proposed Method** |
> |-------------------|---------------|----------------|--------|---------|---------|----------|-----------|---------------------|
> | Allen-Cahn        | 1D Space/Time | Periodic       | 34.99  | 17.65   | 64.81   | 22.29    | 116.72    | 150.88              |
> | DarcyFlow         | 2D Annulus    | Neumann        | 2.37   | 2.76    | 18.67   | 2.18     | 11.88     | 5.73                |
> | (smooth coeff.)   |               | Dirichlet      | 2.08   | 2.09    | 17.64   | 1.64     | 10.87     | 11.87               |
> |                   | 3D Annulus    | Neumann        | 8.1    | 8.5     | 27.6    | 6.9      | 20.8      | 35.9                |
> |                   |               | Dirichlet      | 5.9    | 6.0     | 21.3    | 5.2      | 15.3      | 30.1                |
> | (rough coeff.)    | 2D Box        | Neumann        | 2.4    | 2.4     | 7.7     | 2.1      | 7.7       | 8.4                 |
>
> > **Regarding PDE benchmarks**
>
> We appreciate the reviewer’s comments regarding the limited number of benchmark PDEs in our current experiments. While the selected problems (e.g., Allen-Cahn and Darcy Flow) were chosen to evaluate distinct aspects of our method, such as handling nonlinear dynamics and varying boundary conditions, we recognize that additional examples would further strengthen the evaluation. If accepted, we plan to expand our experiments to include additional PDEs.
>
> > **RBF kernel explanation.**
>
> We appreciate the reviewer's observation regarding Figure 3 and would like to clarify the relationship between kernel smoothness and the ability to approximate sharp transitions. While Figure 3 visualizes the learned RBF kernels, their smooth appearance reflects the inherent properties of individual RBF kernels rather than the overall behavior of the hybrid model.
>
> Specifically, when using Wendland kernels—a compactly supported RBF kernel—the smoothness is localized within the kernel's support region, and the kernel transitions sharply to zero at the boundary of its support. This compact support introduces non-smooth behavior at the edges while maintaining smoothness within the support region. The hybrid model benefits from the locality in the Wendland kernels, which ensures that each kernel focuses on a specific region of the solution, while the neural network component provides the flexibility to adjust to global features. Together, each component enables the hybrid model to balance smooth behavior and sharp transitions effectively.
>
> Further, to directly address this concern, we refer the reviewer to Figure 4, which provides evidence of the HyResPINN's ability to handle sharp transitions. Specifically, the absolute error plots in the second row of Figure 4 illustrate that HyResPINN consistently achieves lower errors near sharp transitions compared to both the standard PINN and the RBF network (RBFNet). These results highlight the hybrid model's effectiveness in capturing sharp features where other methods falter. The bottom row of Figure 4 shows that HyResPINN accurately captures smooth and sharp features in the Allen-Cahn solution across all time steps, aligning closely with the exact solution (green). In contrast, the standard PINN struggles to resolve sharp features, resulting in large errors.

---

> ### Author Response · Authors · 2024-12-04
> **Response to Reviewer ba3x (part 2)**
>
> > **Are the centers and coefficients trained in each RBF-NN?**
>
> In our proposed adaptive RBF kernel approach, the scaling parameters ($\tau_i$) of the Wendland kernel, as well as the kernel weights ($W$ in Equation 12), are trainable model parameters. These parameters are optimized via gradient descent alongside all other network parameters during training.
>
> We thank the reviewers for their careful reading of our manuscript and for pointing out the grammatical and notational errors. In the revised manuscript, we have updated the notation in Equation (4), fixed the grammatical errors, and fixed the reference pointers in the Appendix.

---

### Meta-Review · Area_Chair_9SoU · 2024-12-22

**Metareview:**

The paper describes a new class of physics-informed neural networks that combines a residual block of RBF and neural networks. The approach is quite strange. First of all, for RBF it is known that Random Fourier Features (RFF) can approximate well RBF functions and kernel methods, making the architecture close to a classical neural networks. Other more formal concerns include:
1) Too narrow experiments
2) Not to good writing (for example, Appendix A has broken references and text has not been updated)
3) Questions about the experiments, i.e. training set selection is not fully correct.
Overall, I don't think the proposal is really interesting, even if it shows some improvements in the experiments.

**Additional Comments On Reviewer Discussion:**

A lot of comments have been provided by AtLi who even asked if the text has been updated, but the authors did not do it.

---

### Decision · Program_Chairs · 2025-01-22

Reject